# Predefined-Time Control for Uncertain High-Order Nonlinear Systems with Quantized Input Signal*

1st Lin Zhao
*College of Science*
*Liaoning University of Technology*
Jinzhou, China
zhaolin19991127@163.com

2nd Shuai Sui
*College of Science*
*Liaoning University of Technology*
Jinzhou, China
shuaisui2011@163.com

*Abstract*—The paper focuses on studying an adaptive fuzzy predefined-time tracking control method for a category of uncertain high-order nonlinear systems with input quantization. The considered plants contain unknown nonlinear functions, input quantization, and external disturbances. To handle the difficulties introduced by the uncertain nonlinearities within the original systems, fuzzy logic systems are employed to estimate the unknown nonlinear functions, while power integrator technology is utilized to address the challenge posed by high-order terms. Using the predefined-time Lyapunov stability theory based on the backstepping recursive technique, the system stability analysis is presented, and it is demonstrated that all signals in the closed-loop system remain bounded within the preset time interval.

*Index Terms*—high-order nonlinear systems, backstepping control, predefined-time, quantized control

## I. Introduction

The time-optimal control problem has garnered significant attention recently. To achieve control objective in a constrained time, finite-time control, fixed-time control, and predefined-time control have been widely researched [1]–[3]. A notable limitation of finite-time and fixed-time control strategies is the inability to establish a direct correlation between settling time and tuning parameters. In contrast, predefined-time control shows greater promise because the stability time of system is determined solely by one design parameter, allowing for the presetting of stability time through parameter modification. In [3], a predefined-time output feedback control strategy is formulated for nonlinear systems with event triggering, further advancing practical predefined-time theories. These advantages make the predefined-time control show a extensive practical application prospect in robotic manipulators [4], spacecraft tracking control [5] and other fields.

High-order (p-normal) nonlinear systems have generated significant concern because of their widespread existence and more general models [6]–[9]. Qian and Lin [6] put forward a control scheme for a category of nonlinear high-order systems by devising innovative iterative-type Lyapunov functions, referred to as adding a power integrator. Since then, many significant outcomes have been attained through the

application of this technology, leading to remarkable advancements, including high-order multi-agent systems [7], high-order interconnection systems [8], and high-order stochastic systems [9]. It should be noted that the above articles do not take into account the adaptive fuzzy predefined-time control problem for uncertain high-order nonlinear systems.

There has been considerable interest in the advancement of quantized control systems due to its theoretical and practical significance in the realms of digital control systems and networked control systems. The authors in [10], [11] studied the problems of adaptive backstepping control for uncertain nonlinear systems with quantized input. To achieve the control objective with sufficient accuracy under low communication rates, the authors in [12] utilized a logarithmic quantizer in channels with communication constraints. In [13], hysteresis quantizers were employed to tackle input quantization problem for nonlinear interconnected systems with weak interconnections. It is evident that there is a lack of existing research on predefined time control for high-order systems, particularly considering input quantization and external disturbances. Inspired by the preceding statements, we aim to study the adaptive fuzzy predefined-time quantized control problem for uncertain high-order nonlinear systems. The primary advantages of this article are outlined as below:

(1) Based on the backstepping recursive technology, this paper first considers quantized-input-based predefined time control design problem for uncertain high-order nonlinear systems, specifically addressing the difficulties that occur when the predefined-time control approach is unable to smoothly transition to higher-order systems due to the presence of higher-order terms.

(2) This article proposes an innovative predefined time control strategy that addresses the inability of finite time and fixed time controllers for high-order nonlinear systems [1], [2] to set system convergence time beforehand. The point in time when the system attains stability can be preset through the adjustment of parameters, making the system have good practical application prospects.

(3) Different from the widely adopted logarithmic quantizer [12], the hysteretic quantizer in this article can avoid chattering issue in the process of quantization and optimize communication bandwidth while maintaining control performance.

This work was funded in part by the National Natural Science Foundation of China (Nos. 62333004, 62176111), in part by the Excellent Young Scientists Fund Program(Overseas), and in part by the "Xingliao Talent Plan" for young top talents, under Grant XLYC 2203188.

## II. PRELIMINARIES AND PROBLEM FORMULATIONS

### A. System Formulation and Assumption

Consider uncertain high-order nonlinear systems as below:

$$\begin{aligned}
\dot{x}_s &= x_{s+1}^{\kappa_s} + f_s(x) + \omega_s(t) \\
\dot{x}_m &= q^{\kappa_m}(u(t)) + f_m(x) + \omega_m(t) \\
y &= x_1
\end{aligned} \tag{1}$$

where $s = 1, 2, ..., m$, $\bar{x}_s = [x_1, ..., x_s]^T$, and $x = \bar{x}_m$ is the state vector; $\kappa_s \geq 1$ denote the positive odd numbers; $f_s(x)$ are the uncertain smooth functions. $u(t)$ and $y$ present the control input and system output of the system, respectively. $\omega_s(t)$ are the bounded external disturbances and satisfy $|\omega_s(t)| \leq \omega_s^*$ with $\omega_s^*$ being unknown positive constants.

As stated in [10], a hysteresis quantizer is described as

$$q(u) = \begin{cases}
u_s sgn(u), \frac{u_s}{1+\rho} \leq |u| \leq u_s, \dot{u} < 0, \\
\quad or \ u_s \leq |u| \leq \frac{u_s}{1-\rho}, \dot{u} > 0 \\
u_s(1+\rho) sgn(u), u_s \leq |u| \leq \frac{u_s}{1-\rho}, \dot{u} < 0, \\
\quad or \ \frac{u_s}{1-\rho} \leq |u| \leq \frac{u_s(1+\rho)}{1-\rho}, \dot{u} > 0 \\
0, 0 \leq |u| \leq \frac{u_{min}}{1+\rho}, \dot{u} < 0, \\
\quad or \ \frac{u_{min}}{1+\rho} \leq |u| \leq u_{min}, \dot{u} > 0 \\
q(u(t^-)), \dot{u} = 0
\end{cases} \tag{2}$$

where $u_s = \delta^{1-s} u_{min}(s = 1, 2, \cdots, m)$ and $\rho = \frac{1-\delta}{1+\delta}$ with $u_{min} > 0$ and $0 < \delta < 1$. $q(u)$ locates into the set $U = \{0, \pm u_s, \pm u_s(1+\rho), s = 1, 2, \cdots, m\}$, $u_{min} > 0$ is the size of the dead-zone for $q(u)$, and $\delta$ stands for the measure of quantization density, and $q(u)$ can be divided into two segments as follows:

$$q^{\kappa_m}(u(t)) = h(u) u^{\kappa_m}(t) + \tau(t) \tag{3}$$

where $h(u)$ and $\tau(t)$ satisfy

$$(1-\rho)^{\kappa_m} \leq h(u) \leq (1+\rho)^{\kappa_m}, \ |\tau(t)| \leq u_{min}^{\kappa_m}. \tag{4}$$

*Assumption 1:* [8] Positive odd numbers $\kappa_s$ satisfy

$$\frac{\kappa+1}{\kappa_s} \geq \kappa - \kappa_{s+1} + 1 \tag{5}$$

where $s = 1, 2, \cdots, m-1$ and $\kappa$ satisfies $\kappa = \max\limits_{s=1,...,m}\{\kappa_s\}$.

*Control Objective:* This article aims to propose an adaptive predefined-time tracking controller for high-order nonlinear systems with quantized input signal, such that all the signals in the closed-loop uncertain high-order nonlinear system are bounded and the tracking error can converge to a predefined region near the origin within some predefined time.

### B. Fuzzy Logic Systems

In this article, FLSs will be applied to the modeling of the unknown nonlinear functions. The FLSs can be denoted as

$$y(\tau) = \frac{\sum_{\zeta=1}^{N} \bar{y}_\zeta \prod_{s=1}^{n} \varsigma_{\mathcal{F}_s^\zeta}(\tau_s)}{\sum_{\zeta=1}^{N}\left[\prod_{s=1}^{n} \varsigma_{\mathcal{F}_s^\zeta}(\tau_s)\right]} \tag{6}$$

where $\bar{y}_\zeta = \max_{y \in R} \varsigma_{G^\zeta}(y)$.

Devise the fuzzy basis functions as below:

$$\varphi_\zeta(\tau) = \frac{\prod_{s=1}^{n} \varsigma_{\mathcal{F}_s^\zeta}(\tau_s)}{\sum_{\zeta=1}^{N}\left[\prod_{s=1}^{n} \varsigma_{\mathcal{F}_s^\zeta}(\tau_s)\right]}. \tag{7}$$

Denote $\vartheta = [\bar{y}_1, \bar{y}_2, \cdots, \bar{y}_N]^T = [\vartheta_1, \vartheta_2, \cdots, \vartheta_N]^T$ and $\varphi^T(\tau) = [\varphi_1(\tau), \varphi_2(\tau), \cdots, \varphi_N(\tau)]^T$, then the fuzzy logic system (6) can be reformulated as follows:

$$y(\tau) = \vartheta^T \varphi(\tau). \tag{8}$$

*Lemma 1:* [3] Define a smooth function $\mathcal{H}(\tau)$ on a compact set $\mho$. For $\forall \epsilon > 0$, there exists a vector $\vartheta^*$ guarantee the inequality

$$\sup_{\tau \in \mho}\left|\mathcal{H}(\tau) - \vartheta^{*T}\varphi(\tau)\right| \leq \epsilon \tag{9}$$

where $\vartheta^*$ represents the ideal parameter vector.

*Lemma 2:* [10] If $\Pi(\Xi) = [x_1(\Xi), ..., x_l(\Xi)]^T$ is the fuzzy basis function vector of FLSs and $\Xi = [z_1, ..., z_m]^T$ is the input vector, then we get the inequality

$$\|\varphi(\Xi)\|^2 \leq \|\varphi(\Xi_s)\|^2 \tag{10}$$

where $\Xi_s = [z_1, ..., z_s]^T$, $0 < s \leq m$.

### C. Predefined Time Stability Theory

The dynamic equation are considered as below:

$$\dot{\tau} = \Gamma(\tau) \tag{11}$$

where the origin is an equilibrium point. $\Gamma : \mathcal{R}^n \to \mathcal{R}^n$ is a nonlinear function and $\tau \in \mathcal{R}^n$ is the system state vector.

*Definition 1:* [3] Given positive constant $T_d > 0$ and $\epsilon > 0$, when $t > T_d$, the state trajectory $\tau$ satisfies $\|\tau\| < \epsilon$. Following that, the origin is practically predefined time stable. Moreover, $T_d$ is called the predefined time.

*Lemma 3:* [3] If there exists a Lyapunov function satisfying

$$\dot{V} \leq -\frac{\pi}{rT_d}V^{1+\frac{r}{2}} - \frac{\pi}{rT_d}V^{1-\frac{r}{2}} + b \tag{12}$$

where $0 < r < 1$, $T_d$ and $b$ are positive constants, $V$ denotes practical predefined time stable and $2T_d$ is the predefined time.

## III. ADAPTIVE FUZZY CONTROL DESIGN AND ITS STABILITY ANALYSIS

In this section, an adaptive fuzzy predefined-time tracking controller will be designed based on the coordinate transformations as below:

$$\begin{aligned}
z_1 &= x_1 - y_r \\
z_s &= x_s - \alpha_{s-1}
\end{aligned} \tag{13}$$

where $s = 2, ..., m$; $y_r$ is the reference signal and its m-th derivative are available, continuous and bounded; $\alpha_{s-1}$ is the intermediate control signal, which will be designed later.

**Step 1 :** From (1) and (13), the derivative of $z_1$ can be indicated as

$$\dot{z}_1 = x_2^{\kappa_1} + f_1(x) + \omega_1(t) - \dot{y}_r. \tag{14}$$

Choose the power-based Lyapunov function

$$V_1 = \frac{z_1^{\kappa-\kappa_1+2}}{\kappa-\kappa_1+2} + \frac{1}{2}\tilde{\Theta}_1^2 \tag{15}$$

where $\tilde{\Theta}_1 = \Theta_1^* - \hat{\Theta}_1$, $\hat{\Theta}_1$ is the estimated value of $\Theta_1^*$.

Following that, calculating the derivative of $V_1$, one gets

$$\dot{V}_1 = z_1^{\kappa-\kappa_1+1}\left(x_2^{\kappa_1} + \bar{f}_1(x) + \omega_1(t) - \dot{y}_r\right) \\ - z_1^{2(\kappa-\kappa_1+1)} - \tilde{\Theta}_1\dot{\hat{\Theta}}_1 \tag{16}$$

with $\bar{f}_1(x) = f_1(x) + z_1^{\kappa-\kappa_1+1}$.

The uncertain nonlinear function $f_1$ is comprised in $\bar{f}_1$, thus estimating it with FLS yields

$$\bar{f}_1(x) = \theta_1^{*T}\varphi_1(\Xi_1) + \varepsilon_1(\Xi_1) \tag{17}$$

where $\Xi_1 = [x_1, x_2, ..., x_m]^T$, $|\varepsilon_1(\Xi_1)| \le \varepsilon_1^*$, $\varepsilon_1^* > 0$ is a constant.

By applying Young's inequality and Lemma 2, it shows that

$$z_1^{\kappa-\kappa_1+1}\bar{f}_1(x) \le |z_1|^{\kappa-\kappa_1+1}\left(\|\theta_1^*\|\|\varphi_1(\Xi_1)\| + \varepsilon_1^*\right) \\ \le \frac{1}{2c_1^2}z_1^{2(\kappa-\kappa_1+1)}\Theta_1^*\varphi_1^T(\chi_1)\varphi_1(\chi_1) \tag{18} \\ + \frac{c_1^2}{2} + \frac{1}{2}z_1^{2(\kappa-\kappa_1+1)} + \frac{\varepsilon_1^{*2}}{2}$$

$$z_1^{\kappa-\kappa_1+1}\omega_1 \le \frac{1}{2}z_1^{2(\kappa-\kappa_1+1)} + \frac{\omega_1^{*2}}{2} \tag{19}$$

where $\chi_1 = [x_1]^T$, $\Theta_1^* = \|\theta_1^*\|^2$, and $c_1 > 0$ is a design parameter. $\omega_1^*$ is an unknown positive number that satisfies $\|\omega_1\| \le \omega_1^*$.

Plugging (18) and (19) into (16) results in

$$\dot{V}_1 \le z_1^{\kappa-\kappa_1+1}\left(\frac{1}{2c_1^2}z_1^{\kappa-\kappa_1+1}\hat{\Theta}_1\varphi_1^T(\chi_1)\varphi_1(\chi_1)\right. \\ \left. + x_2^{\kappa_1} - \dot{y}_r + \alpha_1^{\kappa_1} - \alpha_1^{\kappa_1}\right) + \frac{\omega_1^{*2}}{2} + \frac{c_1^2}{2} + \frac{\varepsilon_1^{*2}}{2} \tag{20} \\ + \tilde{\Theta}_1\left(\frac{1}{2c_1^2}z_1^{2(\kappa-\kappa_1+1)}\varphi_1^T(\chi_1)\varphi_1(\chi_1) - \dot{\hat{\Theta}}_1\right).$$

Devise the intermediate control signal $\alpha_1$, and the adaptive update law $\dot{\hat{\Theta}}_1$ as

$$\alpha_1 = -\left(\frac{1}{2c_1^2}z_1^{\kappa-\kappa_1+1}\hat{\Theta}_1\varphi_1^T(\chi_1)\varphi_1(\chi_1) + l_1z_1^{\kappa_1}\right. \\ \left. + \frac{\sigma_1\pi}{rT_d}z_1^{\left(1+\frac{r}{2}\right)(\kappa+1)-(\kappa-\kappa_1+1)} - \dot{y}_r\right)^{\frac{1}{\kappa_1}} \tag{21}$$

$$\dot{\hat{\Theta}}_1 = \frac{1}{2c_1^2}z_1^{2(\kappa-\kappa_1+1)}\varphi_1^T(\chi_1)\varphi_1(\chi_1) \\ - 2\left(\frac{(2-r)m^{\frac{r}{2}}\pi}{4rT_d} + \frac{(2-r)\pi}{2^{2-\frac{r}{2}}rT_d}\right)\hat{\Theta}_1 \tag{22}$$

where $\sigma_1 = (2m)^{\frac{r}{2}}/(\kappa-\kappa_1+2)^{\left(1+\frac{r}{2}\right)}$, $l_1 = (\bar{\sigma}_1\pi)/(rT_d) + 1$, $\bar{\sigma}_1 = 1/(\kappa-\kappa_1+2)^{\left(1-\frac{r}{2}\right)}$ and $\hat{\Theta}_1(0) \ge 0$.

Using Young's inequality, it follows that

$$2\left(\frac{(2-r)m^{\frac{r}{2}}\pi}{4rT_d} + \frac{(2-r)\pi}{2^{2-\frac{r}{2}}rT_d}\right)\tilde{\Theta}_s\hat{\Theta}_s \\ \le -\left(\frac{(2-r)m^{\frac{r}{2}}\pi}{4rT_d} + \frac{(2-r)\pi}{2^{2-\frac{r}{2}}rT_d}\right)\left(\tilde{\Theta}_s^2 - \Theta_s^{*2}\right). \tag{23}$$

where $s = 1, 2, ..., m$.

Combining (20) - (23) yields

$$\dot{V}_1 \le z_1^{\kappa-\kappa_1+1}\left(x_2^{\kappa_1} - \alpha_1^{\kappa_1}\right) - l_1z_1^{\kappa+1} \\ - \frac{\sigma_1\pi}{rT_d}z_1^{\left(1+\frac{r}{2}\right)(\kappa+1)} + \frac{\omega_1^{*2}}{2} + \frac{c_1^2}{2} + \frac{\varepsilon_1^{*2}}{2} \tag{24} \\ - \left(\frac{(2-r)m^{\frac{r}{2}}\pi}{4rT_d} + \frac{(2-r)\pi}{2^{2-\frac{r}{2}}rT_d}\right)\left(\tilde{\Theta}_1^2 - \Theta_1^{*2}\right).$$

According to [8], it can be inferred that for any real-valued functions $\mathfrak{T}$, $\mathfrak{L}$, any positive odd integer $\kappa > 1$ and a given positive constant $\zeta$, one gets

$$|\mathfrak{T}^\kappa - \mathfrak{L}^\kappa| \le \kappa|\mathfrak{T} - \mathfrak{L}|(\mathfrak{T}^{\kappa-1} + \mathfrak{L}^{\kappa-1}) \tag{25}$$

$$|\mathfrak{T} + \mathfrak{L}|^\zeta \le c_\zeta\left(|\mathfrak{T}|^\zeta + |\mathfrak{L}|^\zeta\right) \tag{26}$$

where $c_\zeta = \begin{cases} 1, & \zeta < 1 \\ 2^{\zeta-1}, & \zeta \ge 1 \end{cases}$.

By virtue of (13), (25) and (26), we have

$$z_1^{\kappa-\kappa_1+1}\left(x_2^{\kappa_1} - \alpha_1^{\kappa_1}\right) \\ \le \kappa_1(2^{\kappa_1-1} + 1)|z_1|^{\kappa-\kappa_1+1}|z_2|\alpha_1^{\kappa_1-1} \tag{27} \\ + \kappa_1 2^{\kappa_1-1}|z_1|^{\kappa-\kappa_1+1}|z_2|^{\kappa_1}.$$

According to [7], it can be inferred that $\Psi$, $\mathfrak{J}$, and any positive constants $\tau$, $\hbar$ and $\alpha$, one has

$$|\Psi|^\tau|\mathfrak{J}|^\hbar \le \frac{\tau}{\tau+\hbar}\alpha|\Psi|^{\tau+\hbar} + \frac{\hbar}{\tau+\hbar}\alpha^{-\frac{\tau}{\hbar}}|\mathfrak{J}|^{\tau+\hbar}. \tag{28}$$

From (28), it can be obtained that

$$\kappa_1 2^{\kappa_1-1}|z_1|^{\kappa-\kappa_1+1}|z_2|^{\kappa_1} \\ \le \kappa_1 2^{\kappa_1-1}\frac{\kappa-\kappa_1+1}{\kappa+1}\left(\frac{\kappa+1}{\kappa-\kappa_1+1}\frac{1}{\kappa_1 2^{\kappa_1}}\right)z_1^{\kappa+1} \\ + \kappa_1 2^{\kappa_1-1}\frac{\kappa_1}{\kappa+1}\left(\frac{\kappa+1}{\kappa-\kappa_1+1}\frac{1}{\kappa_1 2^{\kappa_1}}\right)^{-\frac{\kappa-\kappa_1+1}{\kappa_1}}z_2^{\kappa+1} \tag{29} \\ = \frac{1}{2}z_1^{\kappa+1} + \delta_{11}z_2^{\kappa+1}$$

$$\kappa_1(2^{\kappa_1-1} + 1)|z_1|^{\kappa-\kappa_1+1}|z_2|\alpha_1^{\kappa_1-1} \le \frac{1}{2}z_1^{\kappa+1} + \delta_{12}z_2^{\frac{\kappa+1}{\kappa_1}} \tag{30}$$

where $\delta_{11} = \kappa_1 2^{\kappa_1-1}\frac{\kappa_1}{\kappa+1}\left(\frac{\kappa+1}{\kappa-\kappa_1+1}\frac{1}{\kappa_1 2^{\kappa_1}}\right)^{-\frac{\kappa-\kappa_1+1}{\kappa_1}}$ and $\delta_{12} = \frac{\kappa_1^2(2^{\kappa_1-1}+1)}{\kappa+1}\left(\frac{\kappa+1}{\kappa-\kappa_1+1}\frac{1}{\kappa_1(2^{\kappa_1}+2)}\right)^{-\frac{\kappa-\kappa_1+1}{\kappa_1}}\alpha_1^{\frac{(\kappa_1-1)(\kappa+1)}{\kappa_1}}$.

Drawing on (27), (29) and (30), one gets

$$\dot{V}_1 \le -L_1z_1^{\kappa+1} + \delta_{11}z_2^{\kappa+1} + \delta_{12}z_2^{\frac{\kappa+1}{\kappa_1}} \\ - \frac{\sigma_1\pi}{rT_d}z_1^{\left(1+\frac{r}{2}\right)(\kappa+1)} - \frac{\bar{\sigma}_1\pi}{rT_d}z_1^{\left(1-\frac{r}{2}\right)(\kappa+1)} \\ + \frac{\bar{\sigma}_1\pi}{rT_d}z_1^{\left(1-\frac{r}{2}\right)(\kappa+1)} + \frac{\omega_1^{*2}}{2} + \frac{c_1^2}{2} + \frac{\varepsilon_1^{*2}}{2} \tag{31} \\ - \left(\frac{(2-r)m^{\frac{r}{2}}\pi}{4rT_d} + \frac{(2-r)\pi}{2^{2-\frac{r}{2}}rT_d}\right)\left(\tilde{\Theta}_1^2 - \Theta_1^{*2}\right)$$

where $L_1 = l_1 - 1$.

Using the inequality (28) produces

$$z_1^{\left(1-\frac{r}{2}\right)(\kappa+1)} \leq \frac{rT_dL_1}{\bar{\sigma}_1\pi}z_1^{\kappa+1} + \frac{r}{2}\left(\frac{rT_dL_1}{\left(1-\frac{r}{2}\right)\bar{\sigma}_1\pi}\right)^{1-\frac{2}{r}} \quad (32)$$

$$-z_1^{\left(1+\frac{r}{2}\right)(\kappa+1)} \leq -z_1^{\left(1+\frac{r}{2}\right)(\kappa-\kappa_1+2)}$$
$$+ \frac{\kappa_1-1}{\kappa+1}\left(\frac{\kappa+1}{\kappa-\kappa_1+2}\right)^{-\frac{\kappa-\kappa_1+2}{\kappa_1-1}} \quad (33)$$

$$-z_1^{\left(1-\frac{r}{2}\right)(\kappa+1)} \leq -z_1^{\left(1-\frac{r}{2}\right)(\kappa-\kappa_1+2)}$$
$$+ \frac{\kappa_1-1}{\kappa+1}\left(\frac{\kappa+1}{\kappa-\kappa_1+2}\right)^{-\frac{\kappa-\kappa_1+2}{\kappa_1-1}}. \quad (34)$$

By plugging in (32) - (34) into (31), one has

$$\dot{V}_1 \leq -\frac{\sigma_1\pi}{rT_d}z_1^{\left(1+\frac{r}{2}\right)(\kappa-\kappa_1+2)} - \frac{\bar{\sigma}_1\pi}{rT_d}z_1^{\left(1-\frac{r}{2}\right)(\kappa-\kappa_1+2)}$$
$$-\left[\frac{(2-r)m^{\frac{r}{2}}\pi}{4rT_d} + \frac{(2-r)\pi}{2^{2-\frac{r}{2}}rT_d}\right]\tilde{\Theta}_1^2$$
$$+ \delta_{11}z_2^{\kappa+1} + \delta_{12}z_2^{\frac{\kappa+1}{\kappa_1}} + \psi_1 \quad (35)$$

where $\psi_1 = \frac{(\bar{\sigma}_1+\sigma_1)\pi}{rT_d}\left(\frac{\kappa_1-1}{\kappa+1}\right)\left(\frac{\kappa+1}{\kappa-\kappa_1+2}\right)^{-\frac{\kappa-\kappa_1+2}{\kappa_1-1}} +$
$\frac{\bar{\sigma}_1\pi}{2T_d}\left(\frac{rT_dL_1}{\left(1-\frac{r}{2}\right)\bar{\sigma}_1\pi}\right)^{1-\frac{2}{r}} + \left[\frac{(2-r)m^{\frac{r}{2}}\pi}{4rT_d} + \frac{(2-r)\pi}{2^{2-\frac{r}{2}}rT_d}\right]\Theta_1^{*2} + \frac{\omega_1^{*2}}{2} +$
$\frac{c_1^2}{2} + \frac{\varepsilon_1^{*2}}{2}$.

**Step** $s$ $(2 \leq s < m)$ **:** Consider the following equation

$$\dot{z}_s = x_{s+1}^{\kappa_s} + f_s(x) + \omega_s(t) - \dot{\alpha}_{s-1}. \quad (36)$$

Choose the power-based Lyapunov function

$$V_s = V_{s-1} + \frac{z_s^{\kappa-\kappa_s+2}}{\kappa-\kappa_s+2} + \frac{1}{2}\tilde{\Theta}_s^2 \quad (37)$$

where $\tilde{\Theta}_s = \Theta_s^* - \hat{\Theta}_s$, $\hat{\Theta}_s$ is the estimated value of $\Theta_s^*$.

Following that, calculating the derivative of $V_s$, one has

$$\dot{V}_s = \dot{V}_{s-1} + z_s^{\kappa-\kappa_s+1}\left(x_{s+1}^{\kappa_s} + \bar{f}_s(x) + \omega_s(t)\right)$$
$$- z_s^{2(\kappa-\kappa_s+1)} - \tilde{\Theta}_s\dot{\hat{\Theta}}_s \quad (38)$$

with $\bar{f}_s(x) = f_s(x) + z_s^{\kappa-\kappa_s+1} - \dot{\alpha}_{s-1}$. Then, FLS $\theta_s^{*T}\varphi_s(\Xi_s)$ is applied to model $\bar{f}_s(x)$

$$\bar{f}_s(x) = \theta_s^{*T}\varphi_s(\Xi_s) + \varepsilon_s(\Xi_s) \quad (39)$$

where $\Xi_s = [x_1, x_2, ..., x_m, \hat{\Theta}_1, \cdots, \hat{\Theta}_{s-1}]^T$, $|\varepsilon_s(\Xi_s)| \leq \varepsilon_s^*$, and $\varepsilon_s^*$ is a positive constant.

By applying Young's inequality and Lemma 2, it shows that

$$z_s^{\kappa-\kappa_s+1}\bar{f}_s(x) \leq \frac{1}{2c_s^2}z_s^{2(\kappa-\kappa_s+1)}\Theta_s^*\varphi_s^T(\chi_s)\varphi_s(\chi_s)$$
$$+ \frac{1}{2}z_s^{2(\kappa-\kappa_s+1)} + \frac{c_s^2}{2} + \frac{\varepsilon_s^{*2}}{2} \quad (40)$$

$$z_s^{\kappa-\kappa_s+1}\omega_s \leq \frac{1}{2}z_s^{2(\kappa-\kappa_s+1)} + \frac{\omega_s^{*2}}{2} \quad (41)$$

where $X_s = [x_1, x_2, \cdots, x_s]^T$, $\Theta_s^* = \|\theta_s^*\|^2$, and $c_s > 0$ is design parameter. $\omega_s^*$ is an unknown positive number that satisfies $\|\omega_s\| \leq \omega_s^*$.

In step $s - 1$, one gets

$$\dot{V}_{s-1} \leq -\sum_{h=1}^{s-1}\frac{\sigma_h\pi}{rT_d}z_h^{\left(1+\frac{r}{2}\right)(\kappa-\kappa_h+2)} - \sum_{h=1}^{s-1}\frac{\bar{\sigma}_h\pi}{rT_d}z_h^{\left(1-\frac{r}{2}\right)(\kappa-\kappa_h+2)}$$
$$- \sum_{h=1}^{s-1}\left[\frac{(2-r)m^{\frac{r}{2}}\pi}{4rT_d} + \frac{(2-r)\pi}{2^{2-\frac{r}{2}}rT_d}\right]\tilde{\Theta}_h^2$$
$$+ \delta_{(s-1)1}z_s^{\kappa+1} + \delta_{(s-1)2}z_s^{\frac{\kappa+1}{\kappa_{s-1}}} + \psi_{s-1}. \quad (42)$$

Then, combining (38) - (42) yields

$$\dot{V}_s \leq -\sum_{h=1}^{s-1}\frac{\sigma_h\pi}{rT_d}z_h^{\left(1+\frac{r}{2}\right)(\kappa-\kappa_h+2)} - \sum_{h=1}^{s-1}\frac{\bar{\sigma}_h\pi}{rT_d}z_h^{\left(1-\frac{r}{2}\right)(\kappa-\kappa_h+2)}$$
$$- \sum_{h=1}^{s-1}\left[\frac{(2-r)m^{\frac{r}{2}}\pi}{4rT_d} + \frac{(2-r)\pi}{2^{2-\frac{r}{2}}rT_d}\right]\tilde{\Theta}_h^2$$
$$+ \tilde{\Theta}_s\left[\frac{1}{2c_s^2}z_s^{2(\kappa-\kappa_s+1)}\varphi_s^T(\chi_s)\varphi_s(\chi_s) - \dot{\hat{\Theta}}_s\right]$$
$$+ z_s^{\kappa-\kappa_s+1}\left(\frac{1}{2c_s^2}z_s^{\kappa-\kappa_s+1}\hat{\Theta}_s\varphi_s^T(\chi_s)\varphi_s(\chi_s)\right.$$
$$\left. + \delta_{(s-1)1}z_s^{\kappa_s} + \delta_{(s-1)2}z_s^{\bar{\kappa}_s} + x_{s+1}^{\kappa_s} - \alpha_s^{\kappa_s} + \alpha_s^{\kappa_s}\right)$$
$$+ \frac{\omega_s^{*2}}{2} + \frac{c_s^2}{2} + \frac{\varepsilon_s^{*2}}{2} + \psi_{s-1} \quad (43)$$

where $\bar{\kappa}_s = \frac{\kappa+1}{\kappa_{s-1}} - (\kappa - \kappa_s + 1)$ and $\bar{\kappa}_s$ is a nonnegative constant based on Assumption 1.

Devise the intermediate control signal $\alpha_s$, and the adaptive update law $\dot{\hat{\Theta}}_s$ as

$$\alpha_s = -\left(\frac{1}{2c_s^2}z_s^{\kappa-\kappa_s+1}\hat{\Theta}_s\varphi_s^T(\chi_s)\varphi_s(\chi_s)\right.$$
$$+ l_sz_s^{\kappa_s} + \delta_{(s-1)1}z_s^{\kappa_s} + \delta_{(s-1)2}z_s^{\bar{\kappa}_s}$$
$$\left. + \frac{\sigma_s\pi}{rT_d}z_s^{\left(1+\frac{r}{2}\right)(\kappa+1)-(\kappa-\kappa_s+1)}\right)^{\frac{1}{\kappa_s}} \quad (44)$$

$$\dot{\hat{\Theta}}_s = \frac{1}{2c_s^2}z_s^{2(\kappa-\kappa_s+1)}\varphi_s^T(\chi_s)\varphi_s(\chi_s)$$
$$- 2\left[\frac{(2-r)m^{\frac{r}{2}}\pi}{4rT_d} + \frac{(2-r)\pi}{2^{2-\frac{r}{2}}rT_d}\right]\hat{\Theta}_s \quad (45)$$

where $\sigma_s = (2m)^{\frac{r}{2}}/(\kappa-\kappa_s+2)^{\left(1+\frac{r}{2}\right)}$, $l_s = (\bar{\sigma}_s\pi)/(rT_d)+1$, $\bar{\sigma}_s = 1/(\kappa-\kappa_s+2)^{\left(1-\frac{r}{2}\right)}$ and $\hat{\Theta}_s(0) \geq 0$.

Combining (23) and (43) - (45), it can be inferred that

$$
\begin{aligned}
\dot{V}_s \leq & -\sum_{h=1}^{s-1} \frac{\sigma_h \pi}{rT_d} z_h^{\left(1+\frac{r}{2}\right)(\kappa-\kappa_h+2)} - \frac{\sigma_s \pi}{rT_d} z_s^{\left(1+\frac{r}{2}\right)(\kappa+1)} \\
& -\sum_{h=1}^{s-1} \frac{\bar{\sigma}_h \pi}{rT_d} z_h^{\left(1-\frac{r}{2}\right)(\kappa-\kappa_h+2)} + z_s^{\kappa-\kappa_s+1}\left(x_{s+1}^{\kappa_s}-\alpha_s^{\kappa_s}\right) \\
& -\sum_{h=1}^{s}\left[\frac{(2-r)m^{\frac{r}{2}}\pi}{4rT_d} + \frac{(2-r)\pi}{2^{2-\frac{r}{2}}rT_d}\right]\tilde{\Theta}_h^2 \\
& +\left[\frac{(2-r)m^{\frac{r}{2}}\pi}{4rT_d} + \frac{(2-r)\pi}{2^{2-\frac{r}{2}}rT_d}\right]\Theta_s^{*2} - l_s z_s^{\kappa+1} \\
& +\frac{\omega_s^{*2}}{2} + \frac{c_s^2}{2} + \frac{\varepsilon_s^{*2}}{2} + \psi_{s-1}.
\end{aligned}
\tag{46}
$$

The following inequality denoted as (47) bears resemblance to (27)–(30) in the step 1, then:

$$
z_s^{\kappa-\kappa_s+1}\left(x_{s+1}^{\kappa_s}-\alpha_s^{\kappa_s}\right) \leq z_s^{\kappa+1} + \delta_{s1} z_{s+1}^{\kappa+1} + \delta_{s2} z_{s+1}^{\frac{\kappa+1}{\kappa_s}}
\tag{47}
$$

where $\delta_{s1} = \kappa_s 2^{\kappa_s-1}\frac{\kappa_s}{\kappa+1}\left(\frac{\kappa+1}{\kappa-\kappa_s+1}\frac{1}{\kappa_s 2^{\kappa_s}}\right)^{-\frac{\kappa-\kappa_s+1}{\kappa_s}}$, and

$\delta_{s2} = \frac{\kappa_s^2\left(2^{\kappa_s-1}+1\right)}{\kappa+1}\left(\frac{\kappa+1}{\kappa-\kappa_s+1}\frac{1}{\kappa_s(2^{\kappa_s}+2)}\right)^{-\frac{\kappa-\kappa_s+1}{\kappa_s}}\alpha_2^{\frac{(\kappa_s-1)(\kappa+1)}{\kappa_s}}$.

Similar to (32) - (34), the following result holds:

$$
z_s^{\left(1-\frac{r}{2}\right)(\kappa+1)} \leq \frac{rT_d L_s}{\bar{\sigma}_s \pi} z_s^{\kappa+1} + \frac{r}{2}\left(\frac{rT_d L_s}{\left(1-\frac{r}{2}\right)\bar{\sigma}_s \pi}\right)^{1-\frac{2}{r}}
\tag{48}
$$

$$
-z_s^{\left(1+\frac{r}{2}\right)(\kappa+1)} \leq -z_s^{\left(1+\frac{r}{2}\right)(\kappa-\kappa_s+2)}
$$
$$
+\frac{\kappa_s-1}{\kappa+1}\left(\frac{\kappa+1}{\kappa-\kappa_s+2}\right)^{-\frac{\kappa-\kappa_s+2}{\kappa_s-1}}
\tag{49}
$$

$$
-z_s^{\left(1-\frac{r}{2}\right)(\kappa+1)} \leq -z_s^{\left(1-\frac{r}{2}\right)(\kappa-\kappa_s+2)}
$$
$$
+\frac{\kappa_s-1}{\kappa+1}\left(\frac{\kappa+1}{\kappa-\kappa_s+2}\right)^{-\frac{\kappa-\kappa_s+2}{\kappa_s-1}}.
\tag{50}
$$

By plugging in (47) - (50) into (46), one gets

$$
\begin{aligned}
\dot{V}_s \leq & -\sum_{h=1}^{s} \frac{\sigma_h \pi}{rT_d} z_h^{\left(1+\frac{r}{2}\right)(\kappa-\kappa_h+2)} - \sum_{h=1}^{s} \frac{\bar{\sigma}_h \pi}{rT_d} z_h^{\left(1-\frac{r}{2}\right)(\kappa-\kappa_h+2)} \\
& -\sum_{h=1}^{s}\left[\frac{(2-r)m^{\frac{r}{2}}\pi}{4rT_d} + \frac{(2-r)\pi}{2^{2-\frac{r}{2}}rT_d}\right]\tilde{\Theta}_h^2 \\
& +\psi_s + \delta_{s1} z_{s+1}^{\kappa+1} + \delta_{s2} z_{s+1}^{\frac{\kappa+1}{\kappa_s}}
\end{aligned}
\tag{51}
$$

where $\psi_s = \frac{(\bar{\sigma}_s+\sigma_s)\pi}{rT_d}\left(\frac{\kappa_s-1}{\kappa+1}\right)\left(\frac{\kappa+1}{\kappa-\kappa_s+2}\right)^{-\frac{\kappa-\kappa_s+2}{\kappa_s-1}} +$
$\frac{\bar{\sigma}_s \pi}{2T_d}\left(\frac{rT_d L_s}{\left(1-\frac{r}{2}\right)\bar{\sigma}_s \pi}\right)^{1-\frac{2}{r}} + \left[\frac{(2-r)m^{\frac{r}{2}}\pi}{4rT_d} + \frac{(2-r)\pi}{2^{2-\frac{r}{2}}rT_d}\right]\Theta_s^{*2} + \frac{\omega_s^{*2}}{2} +$
$\frac{c_s^2}{2} + \frac{\varepsilon_s^{*2}}{2} + \psi_{s-1}$ and $L_s = l_s - 1$.

**Step $m$ :** In this step, an actual control law $u$ is formulated. From (1) , (3) and (13), the time derivative of $z_m$ is

$$
\dot{z}_m = h(u)u^{\kappa_m}(t) + \tau(t) + f_m(x) + \omega_m(t) - \dot{\alpha}_{m-1}.
\tag{52}
$$

Choose the power-based Lyapunov function

$$
V_m = V_{m-1} + \frac{z_m^{\kappa-\kappa_m+2}}{\kappa-\kappa_m+2} + \frac{(1-\rho)^{\kappa_m}}{2}\tilde{\Theta}_m^2
\tag{53}
$$

where $\tilde{\Theta}_m = \Theta_m^* - \hat{\Theta}_m$, $\hat{\Theta}_m$ is the estimated value of $\Theta_m^*$.

Following that, calculating the derivative of $V_m$, one gets

$$
\begin{aligned}
\dot{V}_m \leq & -\sum_{h=1}^{m-1} \frac{\sigma_h \pi}{rT_d} z_h^{\left(1+\frac{r}{2}\right)(\kappa-\kappa_h+2)} - \sum_{h=1}^{m-1} \frac{\bar{\sigma}_h \pi}{rT_d} z_h^{\left(1-\frac{r}{2}\right)(\kappa-\kappa_h+2)} \\
& -\sum_{h=1}^{m-1}\left[\frac{(2-r)m^{\frac{r}{2}}\pi}{4rT_d} + \frac{(2-r)\pi}{2^{2-\frac{r}{2}}rT_d}\right]\tilde{\Theta}_h^2 - \frac{3}{2}z_m^{2(\kappa-\kappa_m+1)} \\
& +z_m^{\kappa-\kappa_m+1}\Big(h(u)u^{\kappa_m}(t) + \tau(t) \\
& +\omega_m(t) + \bar{f}_m(x)\Big) - (1-\rho)^{\kappa_m}\tilde{\Theta}_m\dot{\hat{\Theta}}_m + \psi_{m-1}
\end{aligned}
\tag{54}
$$

where $\bar{f}_m(x) = f_m(x) + \frac{3}{2}z_m^{\kappa-\kappa_m+1} + \delta_{(m-1)1}z_m^{\bar{\kappa}_m} + \delta_{(m-1)2}z_m^{\bar{\kappa}_m} - \dot{\alpha}_{m-1}$, $\bar{\kappa}_m = \frac{\kappa+1}{\kappa_{m-1}} - (\kappa-\kappa_m+1)$, $\bar{\kappa}_m$ is a nonnegative constant based on Assumption 1,

$$
\begin{aligned}
\delta_{(m-1)1} = & \frac{\kappa_{m-1}^2 2^{\kappa_{m-1}-1}}{\kappa+1} \\
& \times\left[\frac{\kappa+1}{\kappa_{m-1}2^{\kappa_{m-1}}(\kappa-\kappa_{m-1}+1)}\right]^{-\frac{\kappa-\kappa_{m-1}+1}{\kappa_{m-1}}}
\end{aligned}
\tag{55}
$$

$$
\begin{aligned}
\delta_{(m-1)2} = & \frac{\kappa_{m-1}^2\left(2^{\kappa_{m-1}-1}+1\right)}{\kappa+1}\alpha_{m-1}^{\frac{(\kappa_{m-1}-1)(\kappa+1)}{\kappa_{m-1}}} \\
& \times\left[\frac{\kappa+1}{\kappa_{m-1}(\kappa-\kappa_{m-1}+1)(2^{\kappa_{m-1}}+2)}\right]^{-\frac{\kappa-\kappa_{m-1}+1}{\kappa_{m-1}}}.
\end{aligned}
\tag{56}
$$

The nonlinear uncertain function $f_m$ is comprised in $\bar{f}_m$, thus estimating it with FLS yields

$$
\bar{f}_m(x) = \theta_m^{*T}\varphi_m(\Xi_m) + \varepsilon_m(\Xi_m)
\tag{57}
$$

where $\Xi_m = [x_1, x_2, ..., x_m, \hat{\Theta}_1, \cdots, \hat{\Theta}_{m-1}]^T$, $|\varepsilon_m(\Xi_m)| \leq \varepsilon_m^*$, and $\varepsilon_m^*$ is a positive constant.

Utilizing Young's inequality, (3), (4) and (28), one has

$$
z_m^{\kappa-\kappa_m+1}\tau(t) \leq \frac{1}{2}z_m^{2(\kappa-\kappa_m+1)} + \frac{1}{2}u_{min}^{2\kappa_m}
\tag{58}
$$

$$
\begin{aligned}
z_m^{\kappa-\kappa_m+1}\bar{f}_m(x) \leq & \frac{(1-\rho)^{\kappa_m}}{2c_m^2}z_m^{2(\kappa-\kappa_m+1)} \\
& \times \Theta_m^*\varphi_m^T(\chi_m)\varphi_m(\chi_m) \\
& +\frac{1}{2}z_m^{2(\kappa-\kappa_m+1)} + \frac{c_m^2}{2} + \frac{\varepsilon_m^{*2}}{2}
\end{aligned}
\tag{59}
$$

$$
z_m^{\kappa-\kappa_m+1}\omega_m \leq \frac{1}{2}z_m^{2(\kappa-\kappa_m+1)} + \frac{\omega_m^{*2}}{2}
\tag{60}
$$

where $\Theta_m^* = \|\theta_m^*\|^2/(1-\rho)^{\kappa_m}$, $X_m = [x_1, x_2, \cdots, x_m]^T$, $c_m > 0$ is design parameter and $\omega_m^*$ is an unknown positive number that satisfies $\|\omega_m\| \leq \omega_m^*$.

Substituting (58) - (60) into (54) gives

$$
\begin{aligned}
\dot{V}_m \leq & -\sum_{h=1}^{m-1} \frac{\sigma_h \pi}{rT_d} z_h^{\left(1+\frac{r}{2}\right)(\kappa-\kappa_h+2)} - \sum_{h=1}^{m-1} \frac{\bar{\sigma}_h \pi}{rT_d} z_h^{\left(1-\frac{r}{2}\right)(\kappa-\kappa_h+2)} \\
& - \sum_{h=1}^{m-1} \left[ \frac{(2-r)m^{\frac{r}{2}}\pi}{4rT_d} + \frac{(2-r)\pi}{2^{2-\frac{r}{2}}rT_d} \right] \tilde{\Theta}_h^2 \\
& + z_m^{\kappa-\kappa_m+1}\Big( h(u)u^{\kappa_m}(t) + \frac{(1-\rho)^{\kappa_m}}{2c_m^2} \\
& \times z_m^{\kappa-\kappa_m+1}\hat{\Theta}_m \varphi_m^T(\chi_m)\varphi_m(\chi_m) \Big) \\
& + (1-\rho)^{\kappa_m}\tilde{\Theta}_m\Big( -\dot{\hat{\Theta}}_m + \frac{1}{2c_m^2} \\
& \times z_m^{2(\kappa-\kappa_m+1)}\varphi_m^T(\chi_m)\varphi_m(\chi_m) \Big) \\
& + \frac{\omega_m^{*2}}{2} + \frac{c_m^2}{2} + \frac{\varepsilon_m^{*2}}{2} + \frac{1}{2}u_{min}^{2\kappa_m} + \psi_{m-1}.
\end{aligned}
$$
(61)

Utilizing the property of quantization (4), design the actual control function $u$, and the adaptive update law $\hat{\Theta}_m$ as below

$$
\begin{aligned}
u = & -\Bigg( \frac{1}{2c_m^2} z_m^{\kappa-\kappa_m+1}\hat{\Theta}_m\varphi_m^T(\chi_m)\varphi_m(\chi_m) \\
& + l_m z_m^{\kappa_m} + \frac{\sigma_m \pi}{rT_d} z_m^{\left(1+\frac{r}{2}\right)(\kappa+1)-(\kappa-\kappa_m+1)} \Bigg)^{\frac{1}{\kappa_m}}
\end{aligned}
$$
(62)

$$
\begin{aligned}
\dot{\hat{\Theta}}_m = & \frac{1}{2c_m^2} z_m^{2(\kappa-\kappa_m+1)}\varphi_m^T(\chi_m)\varphi_m(\chi_m) \\
& - 2\left( \frac{(2-r)m^{\frac{r}{2}}\pi}{4rT_d} + \frac{(2-r)\pi}{2^{2-\frac{r}{2}}rT_d} \right)\hat{\Theta}_m
\end{aligned}
$$
(63)

where $\sigma_m = (2m)^{\frac{r}{2}}/(\kappa-\kappa_m+2)^{\left(1+\frac{r}{2}\right)}$, $l_m = (\bar{\sigma}_m\pi)/((1-\rho)^{\kappa_m}rT_d)$, and $\bar{\sigma}_m = 1/(\kappa-\kappa_m+2)^{\left(1-\frac{r}{2}\right)}$.

Combining (4), (23) and (61) - (63), it can be inferred that

$$
\begin{aligned}
\dot{V}_m \leq & -\sum_{h=1}^{m-1} \frac{\sigma_h \pi}{rT_d} z_h^{\left(1+\frac{r}{2}\right)(\kappa-\kappa_h+2)} - \sum_{h=1}^{m-1} \frac{\bar{\sigma}_h \pi}{rT_d} z_h^{\left(1-\frac{r}{2}\right)(\kappa-\kappa_h+2)} \\
& - \sum_{h=1}^{m} \left[ \frac{(2-r)m^{\frac{r}{2}}\pi}{4rT_d} + \frac{(2-r)\pi}{2^{2-\frac{r}{2}}rT_d} \right] \tilde{\Theta}_h^2 \\
& + \left[ \frac{(2-r)m^{\frac{r}{2}}\pi}{4rT_d} + \frac{(2-r)\pi}{2^{2-\frac{r}{2}}rT_d} \right] \Theta_m^{*2} \\
& - \frac{(1-\rho)^{\kappa_m}\sigma_m\pi}{rT_d} z_m^{\left(1+\frac{r}{2}\right)(\kappa+1)} - (1-\rho)^{\kappa_m}l_m z_m^{\kappa+1} \\
& - \frac{\bar{\sigma}_m\pi}{rT_d} z_m^{\left(1-\frac{r}{2}\right)(\kappa+1)} + \frac{\bar{\sigma}_m\pi}{rT_d} z_m^{\left(1-\frac{r}{2}\right)(\kappa+1)} \\
& + \frac{\omega_m^{*2}}{2} + \frac{c_m^2}{2} + \frac{\varepsilon_m^{*2}}{2} + \frac{1}{2}u_{min}^{2\kappa_m} + \psi_{m-1}.
\end{aligned}
$$
(64)

Similar to (32) - (34), the following result holds:

$$
\begin{aligned}
& z_m^{\left(1-\frac{r}{2}\right)(\kappa+1)} \\
& \leq \frac{rT_d(1-\rho)^{\kappa_m}l_m}{\bar{\sigma}_m\pi} z_m^{\kappa+1} \\
& \quad + \frac{r}{2}\left( \frac{rT_d(1-\rho)^{\kappa_m}l_m}{\left(1-\frac{r}{2}\right)\bar{\sigma}_m\pi} \right)^{1-\frac{2}{r}} \\
& \quad - (1-\rho)^{\kappa_m} z_m^{\left(1+\frac{r}{2}\right)(\kappa+1)}
\end{aligned}
$$
(65)

$$
\begin{aligned}
& \leq -z_m^{\left(1+\frac{r}{2}\right)(\kappa-\kappa_m+2)} \\
& \quad + \frac{\kappa_m-1}{\kappa+1}\left[ \frac{(\kappa+1)(1-\rho)^{\kappa_m}}{\kappa-\kappa_m+2} \right]^{-\frac{\kappa-\kappa_m+2}{\kappa_m-1}}
\end{aligned}
$$
(66)

$$
\begin{aligned}
& -z_m^{\left(1-\frac{r}{2}\right)(\kappa+1)} \\
& \leq -z_m^{\left(1-\frac{r}{2}\right)(\kappa-\kappa_m+2)} \\
& \quad + \frac{\kappa_m-1}{\kappa+1}\left( \frac{\kappa+1}{\kappa-\kappa_m+2} \right)^{-\frac{\kappa-\kappa_m+2}{\kappa_m-1}}.
\end{aligned}
$$
(67)

By plugging in (65) - (67) into (64), we get

$$
\begin{aligned}
\dot{V}_m \leq & -\sum_{h=1}^{m} \frac{(2m)^{\frac{r}{2}}\pi}{(\kappa-\kappa_h+2)^{\left(1+\frac{r}{2}\right)}rT_d} z_h^{\left(1+\frac{r}{2}\right)(\kappa-\kappa_h+2)} \\
& - \sum_{h=1}^{m} \frac{\pi}{(\kappa-\kappa_h+2)^{\left(1-\frac{r}{2}\right)}rT_d} z_h^{\left(1-\frac{r}{2}\right)(\kappa-\kappa_h+2)} \\
& - \sum_{h=1}^{m} \left[ \frac{(2-r)m^{\frac{r}{2}}\pi}{4rT_d} + \frac{(2-r)\pi}{2^{2-\frac{r}{2}}rT_d} \right] \tilde{\Theta}_h^2 + \psi_m
\end{aligned}
$$
(68)

where $\psi_m = \frac{\sigma_m\pi}{rT_d}\left( \frac{\kappa_m-1}{\kappa+1} \right)\left[ \frac{(\kappa+1)(1-\rho)^{\kappa_m}}{\kappa-\kappa_m+2} \right]^{-\frac{\kappa-\kappa_m+2}{\kappa_m-1}} + \frac{\bar{\sigma}_m\pi}{rT_d}\left( \frac{\kappa_m-1}{\kappa+1} \right)\left( \frac{\kappa+1}{\kappa-\kappa_m+2} \right)^{-\frac{\kappa-\kappa_m+2}{\kappa_m-1}} + \frac{\bar{\sigma}_m\pi}{2T_d}\left( \frac{rT_d(1-\rho)^{\kappa_m}l_m}{\left(1-\frac{r}{2}\right)\bar{\sigma}_m\pi} \right)^{1-\frac{2}{r}} + \left[ \frac{(2-r)m^{\frac{r}{2}}\pi}{4rT_d} + \frac{(2-r)\pi}{2^{2-\frac{r}{2}}rT_d} \right]\Theta_m^{*2} + \frac{\omega_m^{*2}}{2} + \frac{c_m^2}{2} + \frac{\varepsilon_m^{*2}}{2} + \frac{1}{2}u_{min}^{2\kappa_m} + \psi_{m-1}$.

Based on [11], one can infer that there is a constant $\varepsilon^*$ such that $|\Psi| \leq \varepsilon^*$. Applying (28), and let $\alpha = \mathfrak{J} = 1, \tau = 2-\eta, \hbar = \eta$ and $\alpha = 1, \mathfrak{J} = \Psi^2, \tau = 2-\eta, \hbar = \eta$, respectively. Following that, the subsequent inequality holds:

$$
-\frac{2-\eta}{2}\Psi^2 \leq -\left(\Psi^2\right)^{1-\frac{\eta}{2}} + \Gamma_\Psi
$$
(69)

$$
-\frac{2-\eta}{2}\Psi^2 \leq -\left(\Psi^2\right)^{1+\frac{\eta}{2}} + \Pi_\Psi
$$
(70)

where $\Gamma_\Psi = \frac{\eta}{2}, \Pi_\Psi = \frac{\eta}{2}\varepsilon^{*4}$.

From (68), it can be deduced that $\tilde{\Theta}_h^2 (h=1,2,...,m)$ are bounded. Thus, there exist positive constants $\bar{\Theta}_h$ such that $\left|\tilde{\Theta}_h\right| \leq \bar{\Theta}_h$. Following that, applying (69) and (70), one gets

$$
-(2-r)\tilde{\Theta}_h^2 \leq -\left(\tilde{\Theta}_h^2\right)^{1-\frac{r}{2}} - \left(\tilde{\Theta}_h^2\right)^{1+\frac{r}{2}} + \frac{r}{2} + \frac{r}{2}\varepsilon^{*4}.
$$
(71)

From the following inequalities proposed in [3], we have

$$-\sum_{\hbar=1}^{n} |\chi_\hbar|^l \le -\left(\sum_{\hbar=1}^{n} |\chi_\hbar|\right)^l, 0 < l \le 1$$

$$-\sum_{\hbar=1}^{n} |\chi_\hbar|^\beta \le -n^{1-\beta}\left(\sum_{\hbar=1}^{n} |\chi_\hbar|\right)^\beta, 1 < \beta \le \infty \tag{72}$$

where $\chi_\hbar(\hbar = 1, 2, ..., n)$ are real variables.

Combining (68) and (71), and applying (72), one has

$$\dot{V}_m \le -\frac{\pi}{rT_d}\left[\sum_{h=1}^{m} \frac{z_h^{\kappa - \kappa_h + 2}}{\kappa - \kappa_h + 2} + \sum_{h=1}^{m}\left(\frac{1}{2}\tilde{\Theta}_h^2\right)\right]^{1+\frac{r}{2}}$$

$$-\frac{\pi}{rT_d}\left[\sum_{h=1}^{m} \frac{z_h^{\kappa - \kappa_h + 2}}{\kappa - \kappa_h + 2} + \sum_{h=1}^{m}\left(\frac{1}{2}\tilde{\Theta}_h^2\right)\right]^{1-\frac{r}{2}} + \psi^* \tag{73}$$

$$\le -\frac{\pi}{rT_d}V_m^{1+\frac{r}{2}} - \frac{\pi}{rT_d}V_m^{1-\frac{r}{2}} + \psi^*$$

where $\psi^* = \frac{m^{1+\frac{r}{2}}\pi r}{2^{2-\frac{r}{2}}rT_d} + \frac{m\pi r\varepsilon^{*4}}{2^{2+\frac{r}{2}}rT_d} + \psi_m$.

Currently, the adaptive fuzzy predefined-time tracking controller approach is finished, and the major result of this study is described in the next theorem.

*Theorem 1:* For high-order nonlinear system (1) under Assumption 1, adaptive update laws (22), (45), (63), controller (62), and intermediate control signal (21), (44), all signals of the closed-loop system are bounded and tracking errors reach a predefined neighborhood of zero in predefined time.

*Proof:* It is evident from (73) that the derivative of $V_m$ satisfies the form of (12) in Lemma 3. Thus, our proposed control strategy can guarantee that the tracking errors of the high-order nonlinear system (1) to be predefined time stable and for $\forall t > 2T_d$, $V < \frac{rT_d\psi^*}{\pi}$.

## IV. CONCLUSION

This paper addresses the problem of an adaptive fuzzy predefined-time tracking control for a class of uncertain high-order nonlinear systems with quantized input signal, unknown nonlinear functions, and external disturbances. The FLSs are employed to approximate these unknown nonlinear functions. The proposed approach integrates adaptive backstepping control with power integrator design technology to develop a fuzzy adaptive predefined-time control scheme. Additionally, the system stability analysis is presented by using the predefined-time Lyapunov stability theory, demonstrating that the proposed adaptive fuzzy control scheme can ensure that all signals in the closed-loop system are bounded within a predetermined time interval. In this paper, the predefined-time control problem for the system (1) was only considered state feedback. Future work will focus on developing an output-feedback predefined-time control scheme for larger-scale high-order nonlinear systems.

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
