# OpenReview forum: "Predefined-Time Control for Uncertain High-Order Nonlinear Systems with Quantized Input Signal"
_IEEE.org/ICIST/2024/Conference — IEEE ICIST 2024 Conference Submission_

### Official Review · Reviewer_79sq · 2024-08-21
**accept**

**Rating:** 7
**Confidence:** 3

**Review:**

Comment: This paper proposes an adaptive fuzzy predefined-time tracking control method for a category of uncertain high-order nonlinear systems with input quantization The theory is correct and can be accepted after responding the following comments.
(1) More comprehensive literature review is needed to clarify the research gap and research motivation.
(2) Definition 1 may come from some existing references. The authors should label the corresponding references.
(3) In the end of the conclusions, some research directions are suggested to be added.

---

### Official Review · Reviewer_cN4i · 2024-08-23
**Predefined-Time Control for Uncertain High-Order Nonlinear Systems with Quantized Input Signal**

**Rating:** 7
**Confidence:** 2

**Review:**

The paper focuses on studying an adaptive fuzzy predefined-time tracking control method for a category of uncertain high-order nonlinear systems with input quantization. The considered plants contain unknown nonlinear functions, input quantization, and external disturbances. The obtained result is valuable and can be accepted if the following problems can be clarified.
1.	The navigational sentence above (57) may be problematic and should be scrutinized.
2.	There are some grammatical mistakes and typos. Please examine the full text further and revise them.
3.	It can be compared with existing articles to make the innovation point clearer.

---

### Official Review · Reviewer_uYQr · 2024-08-27
**The paper focuses on studying an adaptive fuzzy predefined-time tracking control method for a category of uncertain high-order nonlinear systems with input quantization.**

**Rating:** 7
**Confidence:** 3

**Review:**

a The motivation and background of wide practical use of the theoretic results presented should be clearly emphasized to facilitate the readers.
b There are some grammatical mistakes and typos. Please examine the full text further and revise them.
c The unique features of the proposed approaches and the main advantages of the results over others have to be clearly commented. Also, the disadvantage of the proposed method must be described in conclusion.

---

### Decision · Program_Chairs · 2024-09-06

Accept (Oral)